# Factors Controlling Morphotaxa Distributions of Diptera Chironomidae in Freshwaters

Bruno Rossaro [1], Laura Marziali [2], Matteo Montagna [3,4], Giulia Magoga [1], Silvia Zaupa [5] and Angela Boggero [5,*]

1    Dipartimento di Scienze Agrarie ed Ambientali (DISAA), Università degli Studi di Milano, Via Celoria 2, 20133 Milano, Italy; bruno.rossaro@unimi.it (B.R.); giulia.magoga@unimi.it (G.M.)
2    National Research Council-Water Research Institute (CNR-IRSA), Via del Mulino 19, 20861 Brugherio, Italy; laura.marziali@irsa.cnr.it
3    Dipartimento di Agraria, Università degli Studi di Napoli Federico II, Via Università 100, 80055 Portici, Italy; matteo.montagna@unina.it
4    BAT Center—Interuniversity Center for Studies on Bioinspired Agro-Environmental Technology, University of Napoli Federico II, Via Università 100, 80055 Portici, Italy
5    National Research Council-Water Research Institute (CNR-IRSA), Corso Tonolli 50, 28922 Verbania Pallanza, Italy; silvia.zaupa@irsa.cnr.it
*    Correspondence: angela.boggero@irsa.cnr.it

**Abstract:** Many hydrobiological studies have dealt with the autecology of species within the family Chironomidae and discussed factors affecting species distribution. The aim of the present research is to consider the most important factors affecting chironomid species distribution. Habitat type (lentic, lotic, krenal, rhithral, etc.), water temperature, conductivity, and trophic status are confirmed key factors controlling their assemblage structure. Here, we introduce the term "morphotaxon" as the taxonomic level, intermediate between genus and species, more suitable to describe the ecological responses of Chironomidae. The present uncertainty related to species identification is at the base of the proposal, with the assumption that species belonging to the same morphotaxon have similar ecological needs. In this study, this hypothesis was found to be valid, with few exceptions represented by species-rich genera (e.g., *Tanytarsus*, *Paratanytarsus*). The morphotaxon can be viewed as an interim measure waiting for the implementation of new complementary approaches, such as species identification with molecular methods.

**Keywords:** species; morphospecies; morphotype; freshwater ecology; lotic and lentic habitat; self-organizing maps

## 1. Introduction

Among aquatic insects, the dipteran family Chironomidae, commonly referred to as non-biting midges, is the most abundant and species-diverse insect group found in freshwaters [1]. Despite intense studies [2,3], detailed information about the response of single species of this family to environmental factors (autecology) is still lacking and somewhat conflicting. However, within the Chironomidae, some taxa have been recognized as reliable freshwater quality bioindicators. The taxonomic level needed for ecosystem biomonitoring has been a matter of wide debate: the problem of "taxonomic penetration", that is, if species or species group or genus or family are the best taxonomic category to perform biomonitoring, has been under discussion; the choice of a few indicator species, taxonomic reduction, and definition of functional groups were possible alternative solutions proposed [4].

Taxonomic reduction or the use of taxa at a higher rank than species has often been preferred with arbitrary choices, essentially based on the incapacity to use lower taxonomic levels.

A proposal of taxonomic reduction can be based on the concept of **morphospecies**, defined as "a taxonomic species based wholly on morphological differences from related species". The term was applied to organisms classified in the same species when they appear identical by morphological (anatomical) criteria. This concept was applied to species not reproducing sexually or to species known only from fossils and was also applied to chironomids in ecological studies [5]. In principle, one can expect that larvae of different species but with similar morphological characters should have similar ecological needs. Conversely, one cannot exclude the existence of species not separable based on their morphology but with different ecological requirements [6]. For example, large genera, such as *Cricotopus*, *Chironomus*, *Tanytarsus*, and *Polypedilum*, could include almost identical larvae belonging to different species with different ecological needs. In particular, the genus *Cricotopus*, which includes many species with different ecological preferences [7], was cited as an example to support this statement, but it was emphasized that the same genus *Cricotopus*, after a closer examination, has different larval types that can be separated according to morphological characters although at a first glance the larvae may appear identical [8].

Another proposal of taxonomic reduction is the concept of **morphotype**: it was applied to chironomids to describe taxa separable based on morphological characters but not considered valid species because it was described only on larval or pupal stages [9,10], but this term may be equivocal because it was mainly used in a different sense, i.e., to define each group of different types of individuals of the same species. In this sense, it is a synonym of the more general term **polymorphism**.

If used in the first sense, different **morphotypes** with contrasting ecological needs are known within chironomid genera. Examples are *Chironomus thummi*, *C. plumosus*, and *C. salinarius* type within the genus *Chironomus* [11]; *Tanytarsus lugens* and *T. brundini* type within *Tanytarsus* [12]; and *Polypedilum laetum* and *P. nubeculosum* type within the genus *Polypedilum* [2]. Each type includes different species not separable morphologically.

A third possible term proposed here is **morphotaxon** [13]. It was used to describe taxa that can be separated, for example, on the basis of molecular criteria, but are not separable by their morphology. In this case, the term "morphotaxon" has the same meaning of **OTU**, the well-known Operational Taxonomic Unit used in numerical taxonomy [14], recently widely used in culture-independent microbial ecology studies for defining evolutionary units equivalent to species.

To avoid the difficulties in describing the relations between abiotic factors and chironomids, the **functional feeding groups** have also been used [7]. More recently, the concept of morphological type [15] has been also proposed to separate different clusters of taxa. These clusters were applied to species with different feeding habits or with different head capsule morphology, disregarding any phylogenetic consideration, but often the membership of a taxon is arbitrarily assigned to one of these groups, and the fact that different instars belong to different functional groups is ignored [4].

The use of DNA barcoding and DNA metabarcoding to identify chironomid preimaginal stages have been recently proposed [16], but at present, these methods cannot be considered a substitute to traditional morphological analysis mainly due to the lack of comprehensive reference databases for the molecular identification of the species of this family [13,17].

In the present paper, the term **morphotaxon** is selected as an answer to the problem of taxonomic reduction, and it is used for taxa that are supposed to have similar ecological preferences; they can belong to a single species or to a group of species or to a genus not well separable at the larval stage on the basis of morphological characters. The definition of morphotaxon is based exclusively on stereoscopic and optical microscope examination, and it is justified by long taxonomic tradition and experience from the awareness that morphological analysis does not allow separation of species in these circumstances. A morphotaxon may coincide with a genus, a subgenus, a group of species within a large

genus, or, for some well-recognizable species, a morphotaxon may coincide with a single species (e.g., *Brillia bifida*).

An ambitious attempt to furnish information about autecology of all species present in a country (Netherlands) was performed [2,3,18]. In these books, all the species present in the country, with the exception of the tribe Tanytarsini, were described giving the following preferences:

1. Substrate: mud, silt, sand, gravel, stones, periphyton, filamentous algae, macrophytes, wood (mining taxa), and plants;
2. Habitat: terrestrial, lotic, river, kryal, krenal, rhithral, potamal, lentic, lake, littoral, sublittoral, profundal, fresh, brackish, and marine waters;
3. Tolerance to different environmental factors: eurythermal, frigostenothermal, euryhaline, stenohaline, eurioxybiont, stenoxybiont, alcalophilous, and acidophilous.

This information is of much interest but cannot be used if the species is not identifiable at the larval stage. The definition of morphotaxa, even if approximate, is more realistic, so it is suggested that it be used in applied ecology.

In conclusion, it is here proposed to renounce to a detailed separation of taxa in performing ecological studies because of the high risk of erroneous identifications.

The aim of this paper is to define the most relevant factors driving chironomid assemblage composition, considering the assemblage composed of "morphotaxa" instead of species.

For this purpose, a large database, including samples collected in lentic and lotic habitats in Italy since the 1970s, was analysed.

In the present paper, in some cases, the term "species" will still be used instead of the term "morphotaxon", but one must remember that the taxonomic units treated in this paper are not "species" in a strict sense but "morphotaxa".

## 2. Materials and Methods

A large database, including 35,352 sites as objects with 4 factors (habitat, depth zone, year, season), 171 chironomid species, and 31 environmental variables as attributes, was assembled including samples of larvae collected from several freshwater habitats in Italy since the 1970s.

Larval samples were collected in several lotic habitats with a Surber net; examples are: (1) glacial streams [19] (Aosta Valley, Ortles-Adamello); (2) rivers at lower altitude—Bormida, Brembo, and Lambro (Lombardy, Northern Italy) [20,21]; and (3) Mediterranean streams in Cilento (Campania, Southern Italy) [22]. Samples were also collected in lentic habitats: (1) subalpine lakes in sublittoral zone in 2006 in prealps [23]; (2) South subalpine lakes in littoral, sublittoral, profundal zones in 2005 in prealps [24] with an Ekman grab; and (3) lakes included in the Inhabit project [25] with a Ponar grab. The collection of pupal exuviae with a drift net [26] often accompanied larval samples, but data will not be included in the analysis.

Sites and morphotaxa were filed in a Microsoft ACCESS database. The species counts were stored as individuals per square meters (ind m$^{-2}$) considering the different areas sampled with different sampling tools. The association between different species and morphotaxa is given in Supplementary Materials S1. Pooling different species into a single morphotaxon followed two criteria: (1) a morphotaxon coincides with a genus when all the species included in the genus cannot be separated at the larval stage according to morphological criteria and (2) a morphotaxon includes species belonging to the same morphological group within a genus (not all the species of the genus). In a few cases, single species (e.g., *Brillia bifida*, *B. longifurca*) were used as morphotaxa. The keys to identify larvae and separate morphotaxa were: (1) [27–29] for all subfamilies, (2) [30,31] for Diamesinae and Prodiamesinae, (3) [31,32] for Orthocladiinae, and (4) [33] for Chironominae.

Each record was associated with a habitat type. Lakes were classified according to a simplified version of the CNR-IRSA protocol [34], aggregating alpine lakes belonging to types AL-1, AL-2, AL-7, AL-8, AL-9, and AL-10 in the group ALA; small lakes AL-4, AL-5,

and AL-6 in the group LS; large lakes (AL-3) were recoded as LL; Mediterranean lakes ME-2, ME-3, and ME-4 were aggregated in the group ME; and volcanic lakes ME-7 were recorded as V. Running waters were classified according to the European river zonation system [35,36] in kryal (K), krenal (S), rhithral (R), and potamal (P) habitat. A small group with brackish waters (B) was also included.

A complete list of species and of the habitats included in the analysis is provided in Supplementary Materials S1 and S2. Starting from the whole database, excluding sites without chironomids and morphotaxa present in <100 samples to avoid the inclusion of an excessive number of morphotaxa, which would have made the interpretation of the results too complex, a total of 7965 sites and 82 morphotaxa were available for statistical analysis.

Starting from a site (rows) × morphotaxa (columns) matrix, a self-organizing map (SOM) was trained [37]. SOM was preferred to other ordination or clustering methods, such as correspondence analysis, non-metric multidimensional scaling (NMDS), or agglomerative cluster analysis, because it allows a simple representation of sites and morphotaxa in two dimensions, and it is not affected by outliers [38]; SOM optimizes similarities, NMDS the distances between objects [39]. SOM allows a representation of data in two dimensions, creating a map in which sites are plotted according to their similarity in morphotaxa composition. Sites with similar morphotaxa composition are aggregated in cells. The number of cells is chosen to allow an optimal representation of similarities: the larger the number of cells, the larger the resolution. In the present analysis, SOM was trained clustering data in $6 \times 4 = 24$ cells. The 24 cells were numbered from bottom left (1) to top right (24) in the figures. The relative abundance of each morphotaxon can be figured in the map, representing with different colours the different abundances of a morphotaxon in the cells. In the present analysis, the maps were plotted dividing the abundances into 5 classes represented with 5 colours. Cells with similar morphotaxa composition are plotted close together. The presence of empty cells emphasizes the distance between assemblages in filled cells. A codebook matrix is produced in which the SOM cells are rows and morphotaxa are columns. SOM approaches a k-mean clustering in aggregating sites in units.

SOM analysis can be trained both as an unsupervised (USOM) and a supervised (SSOM) self-organizing map, where, respectively, no or one (or more) dependent variables are included to guide the clustering. Here, both USOM and SSOM were trained. Both analyses worked on an X matrix, with sites as rows and morphotaxa as columns. Several SSOM were trained, each including an additional Y matrix with only one column, representing a different factor or environmental variable. The Y matrices included were factors (habitat, season, year, depth zone) or quantitative variables (altitude, source distance of sampled site, current velocity, water temperature, conductivity, dissolved oxygen, oxygen % saturation, total phosphorous, ammonium); the quantitative variables were divided into 5 discrete classes before inclusion in the analysis.

SOM map homogeneity was measured with quantization error (QE). High QE means a more heterogeneous map; conversely, low QE means a more homogenous map [40]. SOM produces a matrix of codebooks, with the 24 cells as rows and the taxa as columns. In the SSOM, the codebook matrix includes additional columns represented by the levels of a factor or the environmental variable.

The codebook matrix obtained from USOM was submitted to a correspondence analysis to have a two-dimensional plot of the 24 cells of the codebook matrix in the plan of the first two principal axes. The scores of the morphotaxa were also added in the biplot to see the relations between morphotaxa and the 24 cells.

All data analyses were carried out in the R environment, using the packages *vegan* [41] and *kohonen* [39]; the function *som* was used for USOM and the function *xyf* for SSOM, with all default options.

## 3. Results

USOM and each SSOM produced codebook matrices, and the corresponding maps are deposited at the University of Milan. The number of samples included and quantization error of each SOM are in Table 1.

**Table 1.** Quantization error of the SOM.

| Dependent Variable | No. Sites | Quantisation Error |
| --- | --- | --- |
| Unsupervised | 7965 | 55.78 |
| Habitat | 7965 | 0.56 |
| Depth zone | 7965 | 0.34 |
| Year | 7965 | 0.51 |
| Season | 7965 | 0.51 |
| Temperature | 3573 | 21.89 |
| Conductivity | 1619 | 65.91 |
| Total phosphorous | 1319 | 73.07 |
| Altitude | 7948 | 61.78 |
| Source distance | 7244 | 62.93 |
| $O_2$ concentration | 2788 | 8.20 |
| $O_2$ % saturation | 2453 | 56.12 |
| Ammonium | 1039 | 59.60 |
| Current velocity | 136 | 2.85 |

The USOM analysis allowed the aggregation of sites exclusively on the basis of similar morphotaxa composition. The different factors (habitat, depth zone, sampling year, season) were not included in the analysis but were passively included in the maps only after the ordination of morphotaxa. To simplify the representation of factors in the maps, years were aggregated into decades from 1970 to 2010, and seasons were generated aggregating months: March, April, and May coded as spring; June, July, and August as summer; September, October, and November as autumn; and December, January, and February as winter. Depth zones considered were littoral (Lit), sublittoral (Sub), and profundal (Pro); running water sites were coded as littoral. Habitat types considered were kryal (K), krenal (S), rhithral (R), potamal (P), brackish waters (B), small (LS) and large lakes (LL), Alpine lakes (ALA), Mediterranean lakes (ME), and volcanic lakes (V).

Morphotaxa clusters obtained with USOM allowed an ordering of morphotaxa in groups in partial agreement with habitat types [34–36] even if habitat types were not always well separated in different cells; in particular, sites belonging to V, ME, LS, and LL were often included together in the same cell. In addition, sites belonging to K, S, and R were often included in the same cell. In Figure 1, each cell is tentatively assigned to one or more habitat types considering the prevalent number of sites assigned to different habitats.

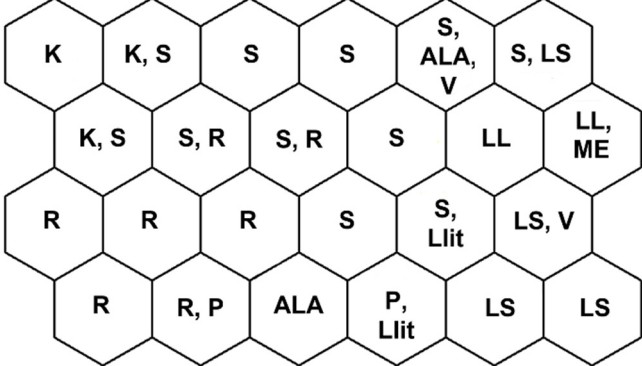

**Figure 1.** Unsupervised map of sites clustered by habitat. Habitat types considered are: kryal (K), krenal (S), rhithral (R), potamal (P), small (LS) and large (LL) lakes, Alpine lakes (ALA), Mediterranean (ME), and volcanic (V) lakes and lake-vegetated littorals (Llit).

The cold stenothermal taxa living in kryal zone or in cold mountain springs, such as *Parorthocladius nudipennis* and *Chaetocladius*, were clustered in cell 19 (Figure 2). *Diamesa bertrami*, *Eukiefferiella brevicalcar*, and *E. minor* were clustered in cells 19 and 20 (Figure 2), *D. latitarsis* and *D. zernyi* were clustered in cells 13, 19, and 20; *D. dampfi*, *Paratrichocladius skirwithensis*, and *Pseudodiamesa* inhabitants of cold mountain springs were aggregated in cells 13 and 19 (Figure 2).

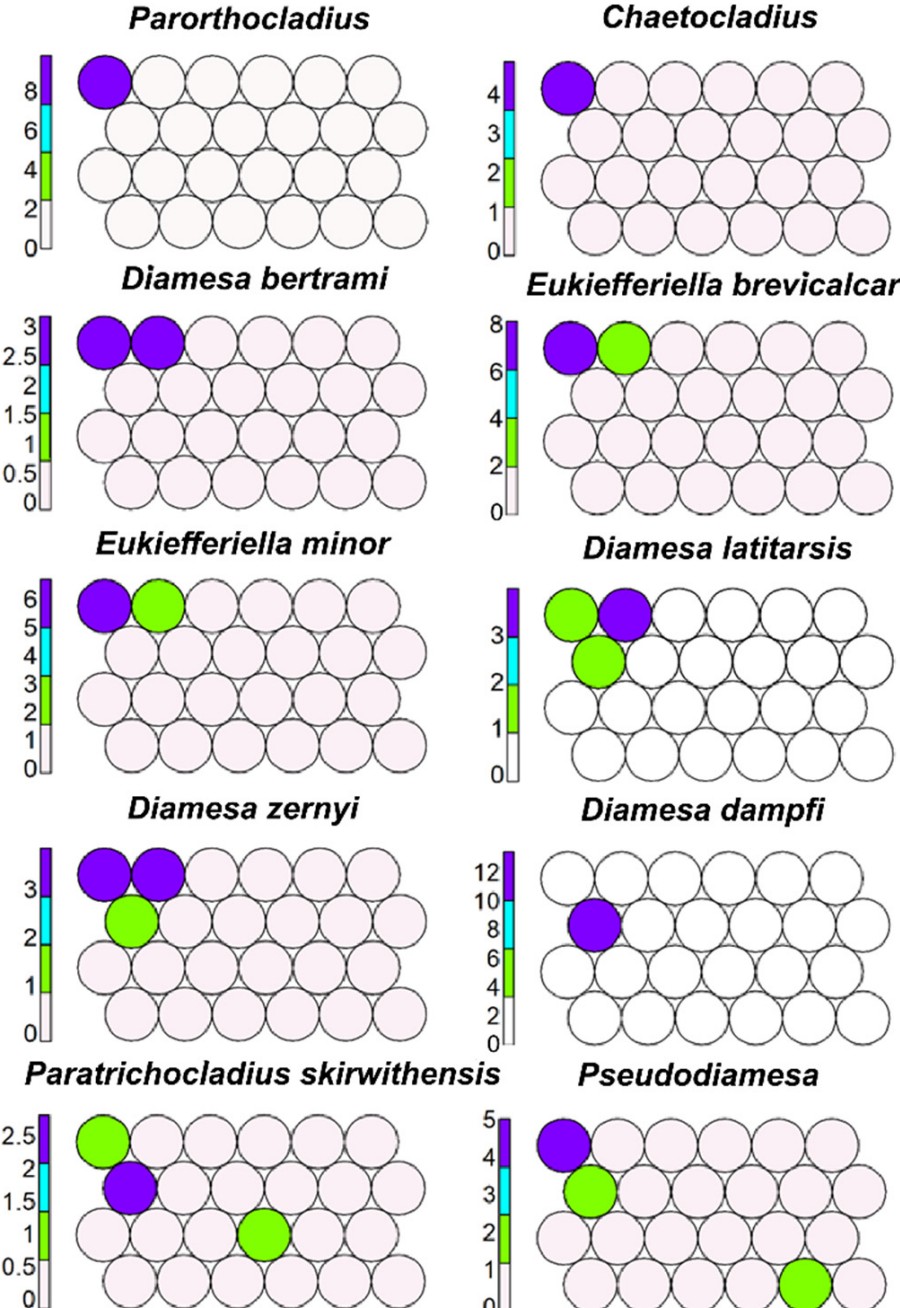

**Figure 2.** Unsupervised map of ten morphotaxa characterizing kryal and cold krenal; the codes and the species associated to morphotaxa are in Supplementary Materials S1 and S2.

Krenophilous morphotaxa, such as *B. bifida* and *B. longifurca*, *Micropsectra atrofasciata*, and *Thienemanniella*, were scattered in several cells (10, 14–16, 21–23) but were mostly in cells 14 and 22 (Figure 3).

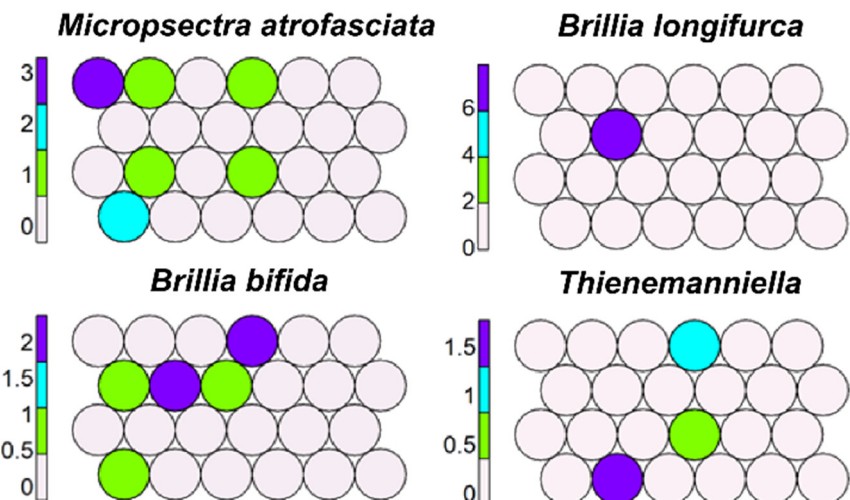

**Figure 3.** Unsupervised map of four morphotaxa characterizing krenal.

Many morphotaxa, such as *Euorthocladius*, *Tvetenia*, *Synorthocladius semivirens*, and *Eukiefferiella claripennis*, were present in several cells characterizing rhithral (cells 1, 7–9, 14–15) (Figure 4). Other morphotaxa, such as *Conchapelopia pallidula* and *Uresipedilum*, were widespread but preferred rhithral (Figure 4). Other morphotaxa mapped in rhithral were *Paracricotopus*, *Paratrichocladius rufiventris*, *Nanocladius*, *Orthocladius* spp., *Rheocricotopus*, and *Polypedilum*.

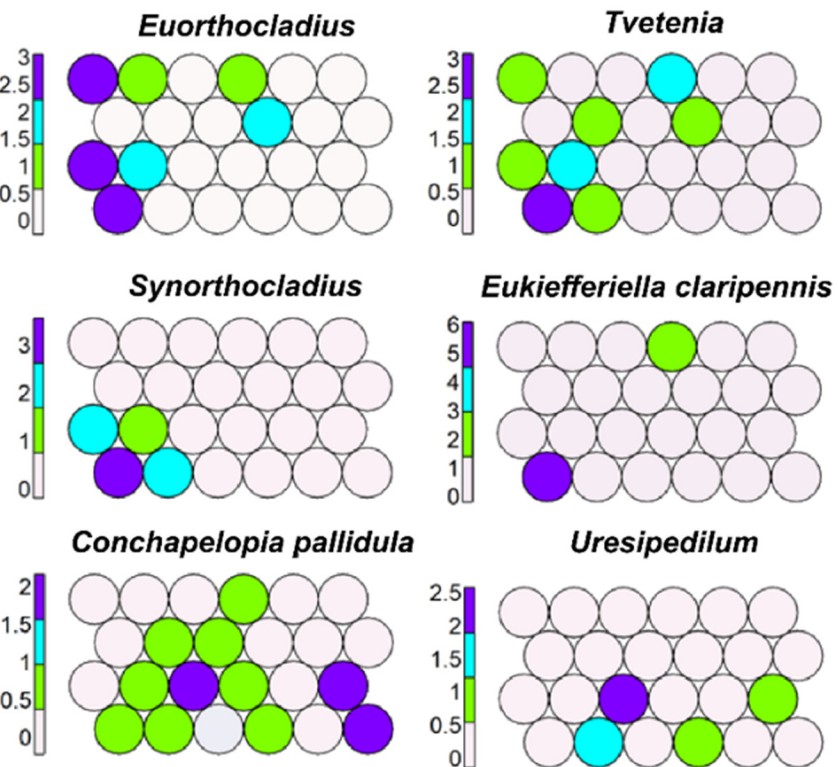

**Figure 4.** Unsupervised map of six morphotaxa characterizing rhithral (upper sector).

*Sympotthastia*, *Orthocladius rubicundus*, *Cricotopus tremulus*, *C. bicinctus* (Figure 5) were mapped in cells representing lower rhithral zone; *Potthastia*, *Rheotanytarsus*, *Eukiefferiella ilkleiensis*, and *Rheopelopia* (Figure 5) were mapped in cells degrading from rhithral to potamal.

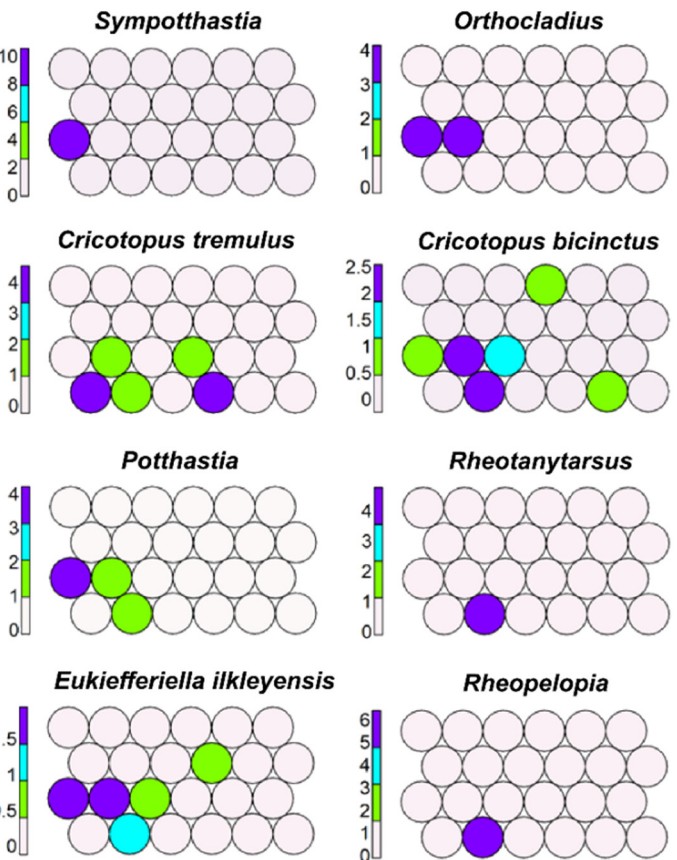

**Figure 5.** Unsupervised map of another eight morphotaxa characterizing rhithral (lower sector).

Taxa associated with aquatic vegetation prevail in cells 4 and 12 (Figure 6): *Pentapedilum sordens*, *Isocladius sylvestris*, *Endochironomus*, *Glyptotendipes*, *Parachironomus gracilior*, and *Microtendipes*.

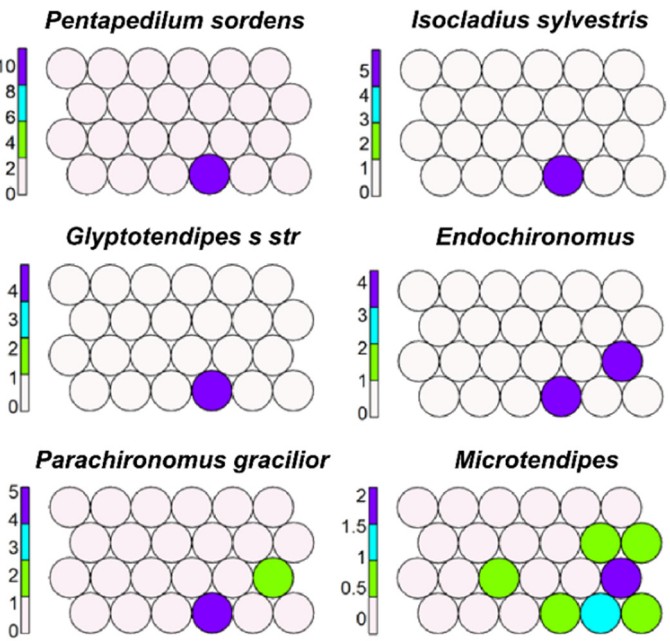

**Figure 6.** Unsupervised map of six morphotaxa associated with vegetation.

*Cryptochironomus, Dicrotendipes, Cladotanytarsus, Demicryptochironomus, Stempellina,* and *Cladopelma lateralis* prevail in cell six and in neighbouring ones (Figure 7); *Pagastiella* and *Parakiefferiella* are also mapped here but occur also in small lakes (LS).

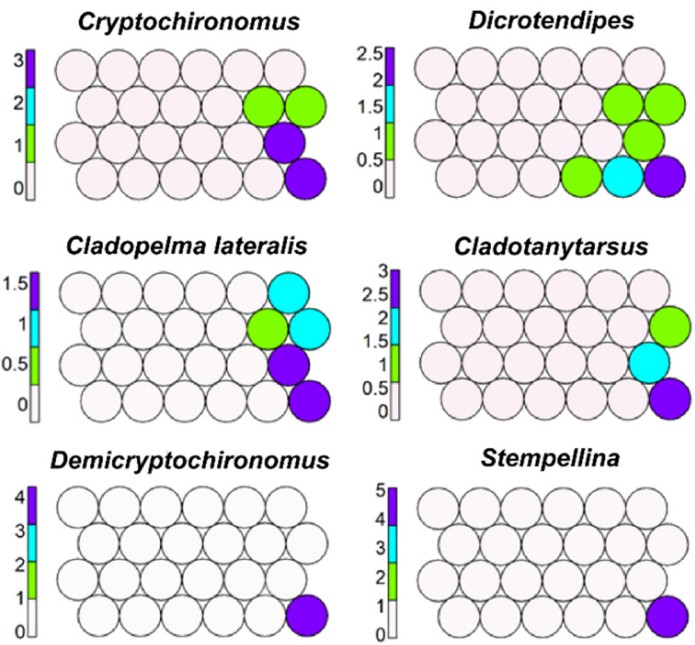

**Figure 7.** Unsupervised map of six morphotaxa characterizing lake sublittorals.

*Chironomus anthracinus, C. plumosus, Microchironomus, Stictochironomus, Prodiamesa olivacea,* and *Micropsectra radialis* prevail in cells 11, 12, and 18 (Figure 8) and usually characterize large lakes (LL), but the last two occur also in cold springs.

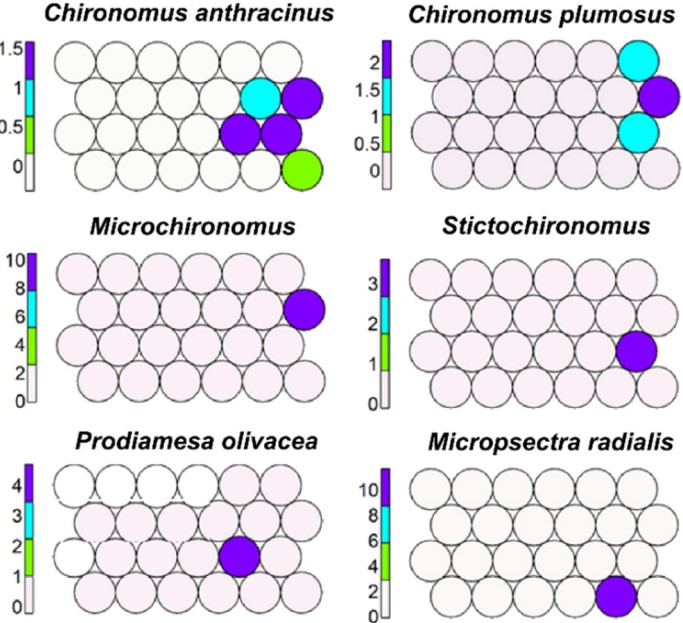

**Figure 8.** Unsupervised map of six morphotaxa characterizing lake profundals.

*Tanypus* and *Paratanytarsus* are present in several cells (Figure 9); *Parataytarsus* includes different morphotaxa that probably live in several habitats, and this explains their dispersed distribution.

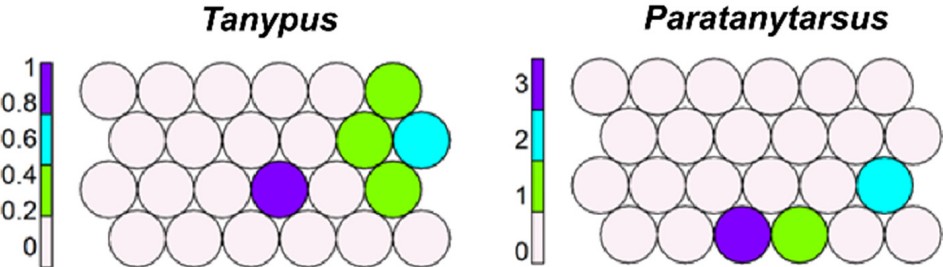

**Figure 9.** Unsupervised map of other two ubiquitous morphotaxa.

*Macropelopia*, *Heterotrissocladius*, *Corynoneura*, and *Zavrelimyia* occupy cell 3 (Figure 10); these taxa are typical of Alpine lakes (ALA), but *Macropelopia* occur also in other habitats (springs, lake profundals), and the morphotype includes different morphotaxa with different preferences; this again explains its distribution in several cells.

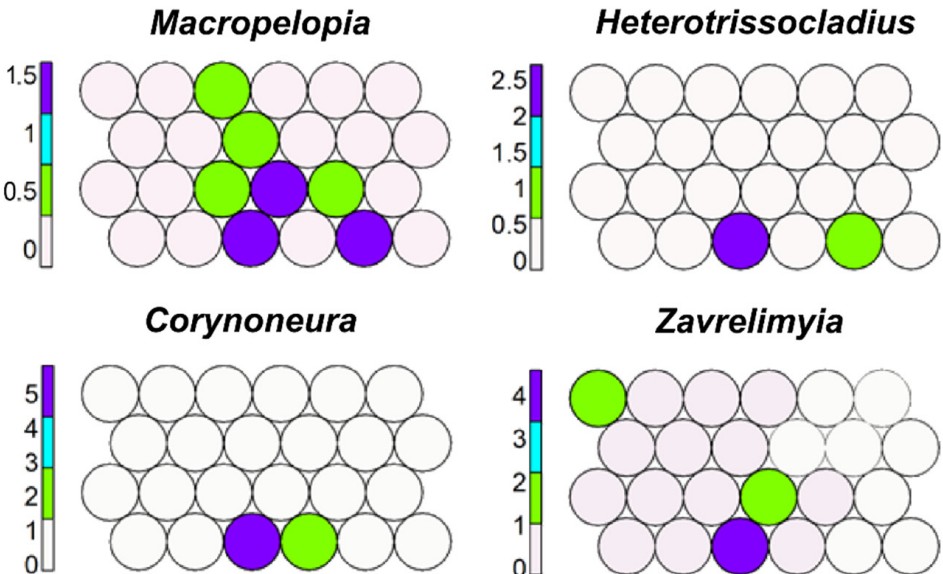

**Figure 10.** Unsupervised map of four morphotaxa characterizing Alpine lakes and cold springs.

A summary of the results was produced by performing a PCA of the codebook matrix produced by USOM, with 82 morphotypes and 24 cells, to plot the morphotaxa scores in the first two axis planes (Figure 11), allowing a summary overview of the association of the morphotypes. Morphotaxa living in cold glacial streams (kryal) or, more generally, in cold alpine streams are plotted in the lower right corner of the map. All *Diamesa* morphotaxa and some *Eukiefferiella* morphotaxa (*E. minor*, *E. brevicalcar*) are plotted here with the cold-stenothermal *Mesorthocladius*, *Chaetocladius*, and *P. nudipennis*.

Morphotaxa living in lower-altitude running waters (rhithral) are plotted in the upper part of the graph. Here, *Paracricotopus*, *P. rufiventris*, *Nanocladius*, *Orthocladius decoratus*, and *Synorthocladius* are found (Figure 11); all these morphotaxa are reophilous.

By moving from the upper part to the middle-left part of the plot, it is possible to find morphotaxa living in a transition zone between rhithral and potamal (hyporhithral, epipotamal), such as *Sympotthastia*, *O. rubicundus*, *Potthastia*, *Rheotanytarsus*, *C. bicinctus*, and *Cricotopus trifascia*.

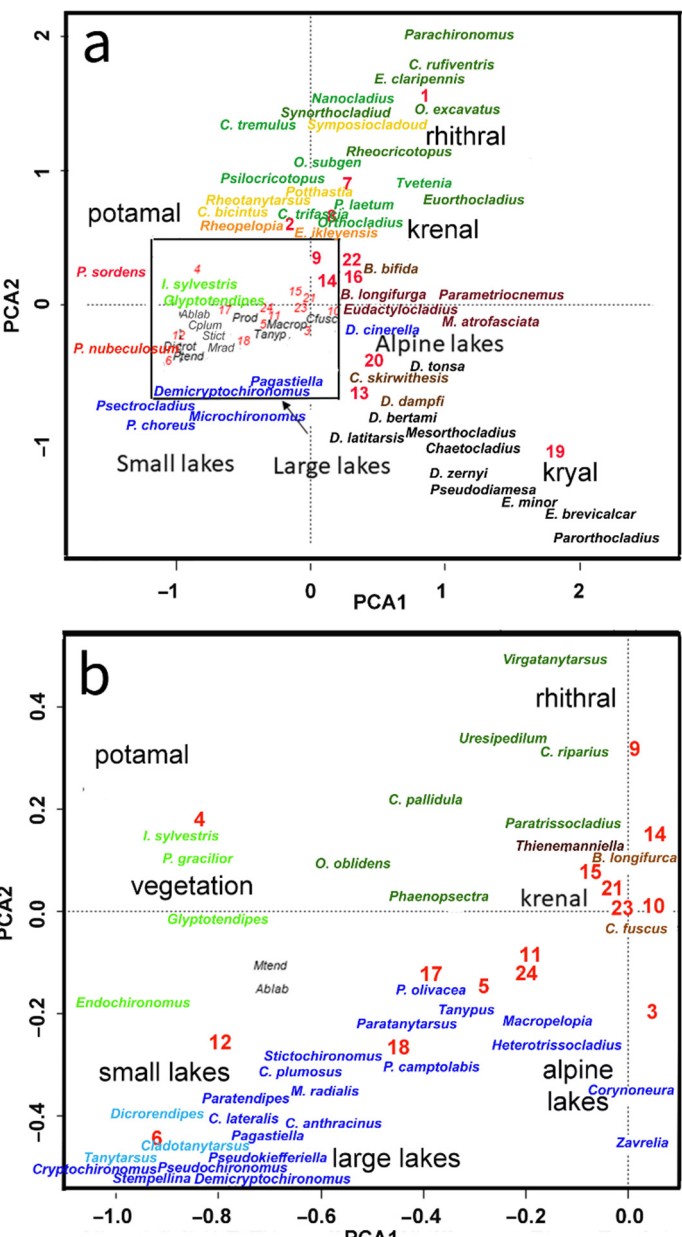

**Figure 11.** Results of PCA analysis: (**a**) PCA map of codebooks; (**b**) detail of map, biplot of morphotaxa, and codebook cells according to the first two axes; morphotaxa labels are coloured according to supposed preferred habitat. Grey, kryal; dark brown, krenal; green, vegetation; dark green, rhithral; dark yellow, epipotamal; red, potamal; blue, lakes.

Moving to the lower left part of the graph, we encounter morphotaxa living along lake littorals, such as *P. nubeculosum*, *Procladius*, *Psectrocladius*, *Microchironomus*, and *Cryptochironomus*. Morphotaxa characterizing large lakes, such as *P. olivacea*, *M. radialis*, *C. anthracinus*, and *C. plumosus*, occur in the centre of the plot (Figure 11).

Environmental data were not available for many sampling sites, so several SSOM were carried out including only the sites for which the selected environmental variable values were available. The quantitative variables were categorized into five classes to allow a clearer representation of the taxa response. In the figures, the highest values of the variables increasing with water quality ($O_2$ concentration and $O_2$ saturation) are in blue and the lowest values in red; the reverse is for the variables increasing with anthropogenic stress ($TP$, $NH_4$), the highest values being in red and the lowest in blue.

Supervised SSOM allowed mapping together sites that jointly have similar values of selected environmental variables and similar morphotaxa composition.

SSOM with habitat as the dependent variable obviously allowed a better separation of morphotaxa according to habitat (Figure 12). It was confirmed that groups of morphotaxa characterizing kryal, krenal, rhithral, and potamal were separated from morphotaxa with lentic preferences, but it was also evident that some morphotaxa (e.g., *Tvetenia, Phaenopsectra*) were not restricted to a single habitat only. For example, *P. laetum,* characterizing rhithral, was also present in brackish waters (Chia bay, Sardinia) (Figure 12); this was not clear in USOM, where a cell joining brackish water sites was not evident.

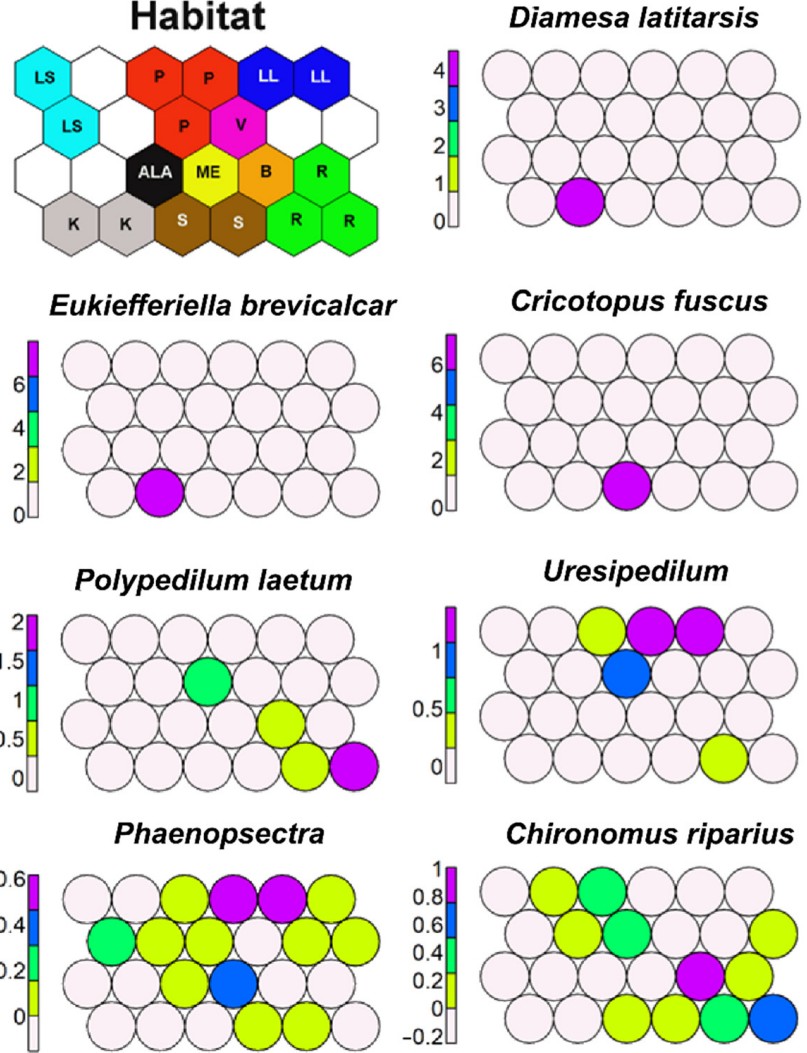

**Figure 12.** Supervised map of sites clustered using habitat as external factor. Habitat types considered are: kryal (K, grey), krenal (S, brown), rhithral (R, green), potamal (P, red), small (LS, azure) and large lakes (LL, blue), Alpine lakes (ALA, black), Mediterranean lakes (ME, yellow), and volcanic lakes (V, magenta), brackish waters (B, orange), and supervised map of seven morphotaxa, using habitat as external factor.

Some taxa, such as *C. anthracinus, Microtendipes, Dicrotendipes,* and *Stictochironomus,* prevailed in lakes, while other morphotaxa, such as *P. choreus, P. olivacea, Macropelopia,* and *Zavrelimyia,* were widespread in several habitats (Figure 13).

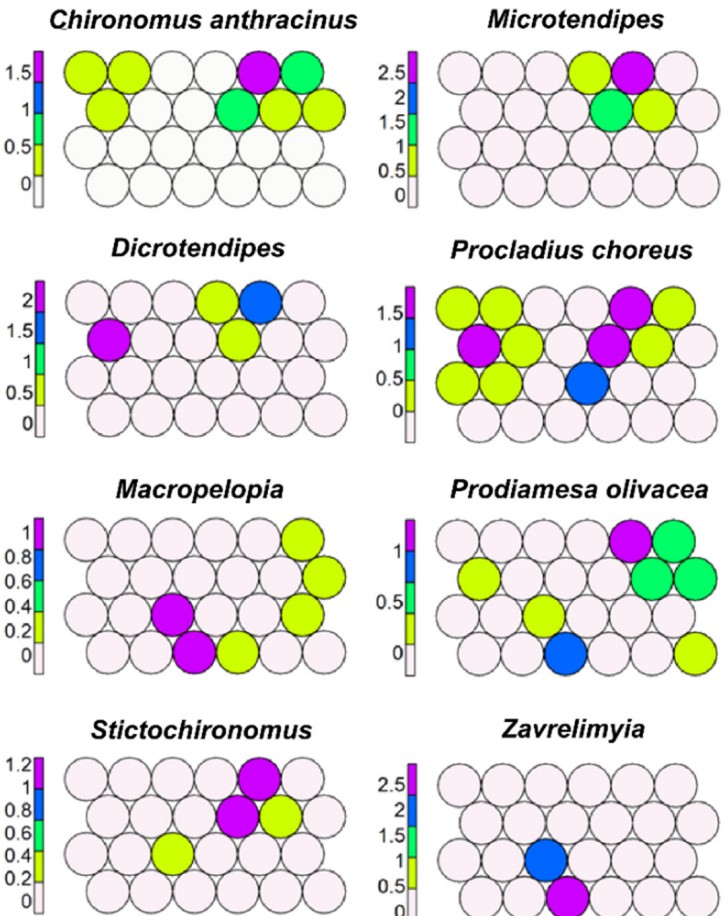

**Figure 13.** Supervised map of another eight morphotaxa using habitat as external factor.

SSOM carried out using depth zone as factor well separated running waters and lake littorals from lake sublittoral and profundal zones (Figure 14). No morphotaxon showed to be restricted to lake profundal or sublittorals, while many morphotaxa were restricted to littoral zone. Only *P. choreus*, *C. anthracinus*, and *P. olivacea* were present in the profundal zone even if not exclusive to it (Figure 14).

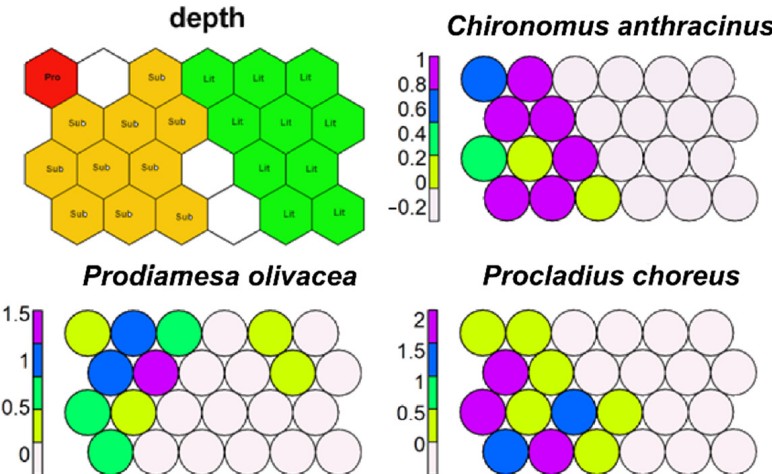

**Figure 14.** Supervised map of sites and three morphotaxa clustered using depth zone as external factor. Running waters and lake littorals (lit, green), lake sublittorals (sub, orange), and lake profundals (pro, red).

SSOM well separated some morphotaxa according to water temperature, but many morphotaxa appeared to respond to a large temperature range (Figure 15).

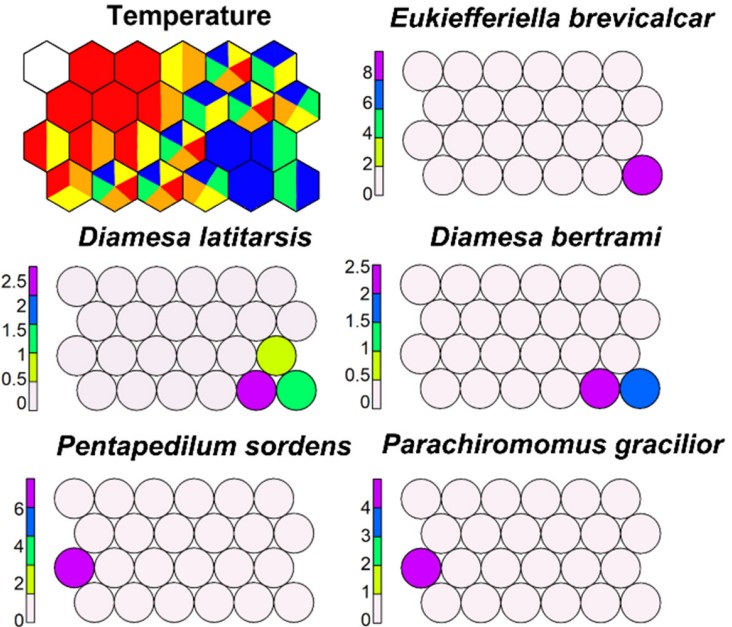

**Figure 15.** Supervised map of sites and five morphotaxa clustered using temperature (in °C) as external variable. Blue, <7.5 °C; green, 7.5–8.5 °C; yellow, 8.5–12 °C; orange, 12–14 °C; red, >14 °C.

All morphotaxa included in the genus *Diamesa* were plotted in the bottom right part of the map in the cells with the lowest water temperature. Few morphotaxa were restricted to high temperatures, such as *P. gracilior* and *P. sordens*.

SSOM carried out using conductivity as an external variable showed that some morphotaxa clearly preferred high-conductivity waters (*C. pallidula*, *C. bicinctus*, *Rheocricotopus*), while others, such as *Mesorthocladius*, *D. latitarsis*, and *D. bertrami*, preferred low-conductivity waters (Figure 16). Figures not included here are available at the University of Milan.

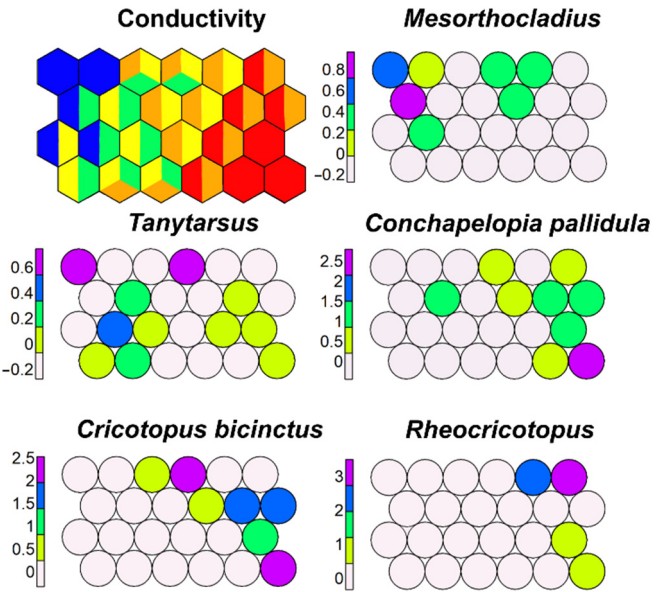

**Figure 16.** Supervised map of sites and five morphotaxa clustered using conductivity (in μS cm$^{-1}$) as external variable. Blue, <100 μS cm$^{-1}$; green, 100–200 μS cm$^{-1}$; yellow, 200–300 μS cm$^{-1}$; orange, 300–400 μS cm$^{-1}$; red, >400 μS cm$^{-1}$.

Sometimes (e.g., *Tanytarsus*), the wide tolerance is because different morphotaxa are included in the same morphotype.

SSOM carried out with TP as the external variable allowed a separation of morphotaxa indicating different trophic status: *C. anthracinus* colonized waters less eutrophic than *C. plumosus*, but *C. thummi* type was the group most tolerant to high TP; *Heterotrissocladius*, *Pagastiella*, *Corynoneura*, and *Psectrocladius* preferred sites with low TP (Figure 17). Figures not included here are available at the University of Milan.

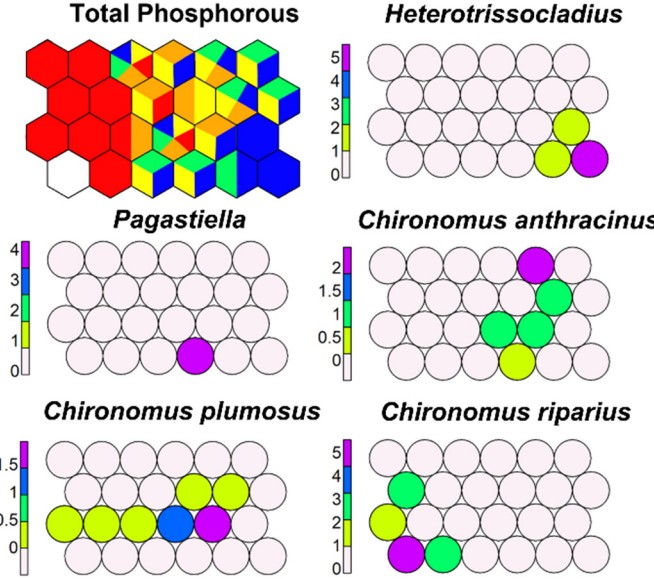

**Figure 17.** Supervised map of sites and five morphotaxa clustered using TP (in µg L$^{-1}$) as external variable. Blue, <10 µg L$^{-1}$; green, 10–20 µg L$^{-1}$; yellow, 20–30 µg L$^{-1}$; orange, 30–50 µg L$^{-1}$; red, >50 µg L$^{-1}$.

SSOM with NH$_4$ as external variable emphasized again a wide tolerance of many morphotaxa, such as *M. atrofasciatata*, *Tanypus*, *C. bicinctus*, and *C. plumosus*, while a few morphotaxa were restricted to cells with low NH$_4$ concentrations (*Demicryptochironomus*) (Figure 18).

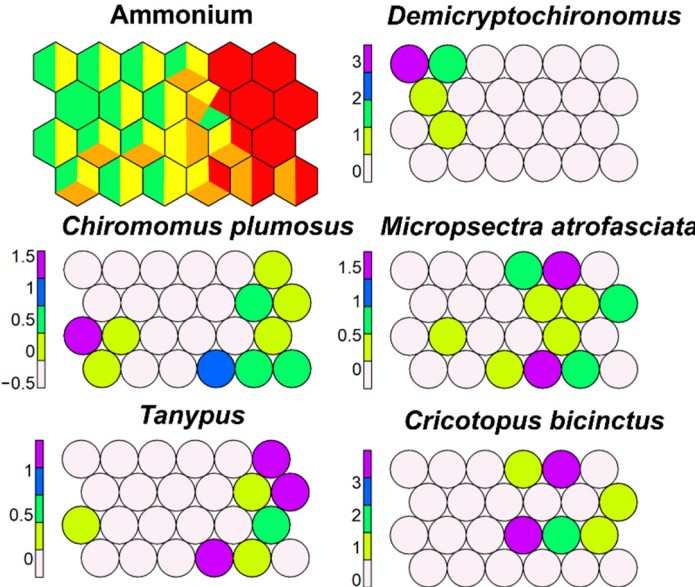

**Figure 18.** Supervised map of sites and five morphotaxa clustered using NH$_4$ (in µg L$^{-1}$) as external variable. Green, 1–20 µg L$^{-1}$; yellow, 20–100 µg L$^{-1}$; orange, 100–250 µg L$^{-1}$; red, >250 µg L$^{-1}$.

Detailed results of all the SSOM for other factors and environmental variables are deposited at the University of Milan. Only a short comment is given here for SSOM trained with other factors/variables. The data available are not well suited to analyse the effect of single factors, so the results obtained need to be confirmed with a larger dataset.

SSOM trained using season as external factor (Figure 19) must be interpreted considering that larval samples include only the fourth instar stage because the first three stages are generally not included in counting as non-identifiable. Most morphotaxa prevail in spring (e.g., *Paratrissocladius*), but some morphotaxa have their maximum growth in summer, in particular the ones living at high altitude. Morphotaxa living at high altitudes, such as *Diamesa latitarsis*, were collected both in summer and autumn.

SSOM including year as dependent factor did not highlight substantial differences in morphotaxa assemblages in different years (Figure 19) and are not reported here.

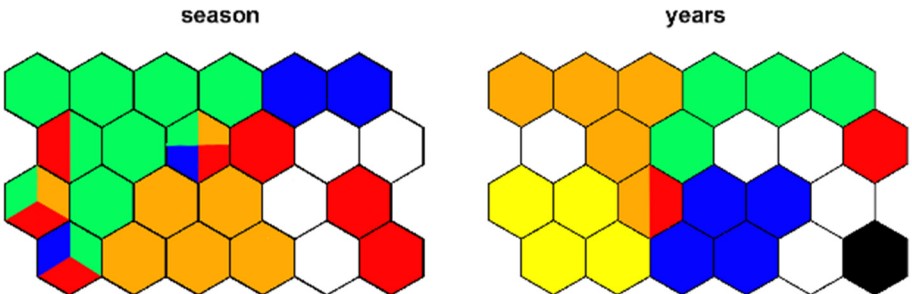

**Figure 19.** Supervised map of sites clustered, respectively, using season (green, spring; red, summer; orange, autumn; blue, winter) and year as external variable.

SSOM with altitude as a dependent variable highlighted that most morphotaxa have no preference for altitude; only a few morphotaxa have maximum growth at high altitude, such as most *Diamesa* morphotaxa, and few at low altitudes, such as *P. gracilior* (Figure 20). Source distance has a meaning only for running-water morphotaxa: SSOM highlighted that most Chironomini prefer stations far from the source, while Diamesini prefer stations near the source, and Orthocladiini do not show clear preferences (Figure 20).

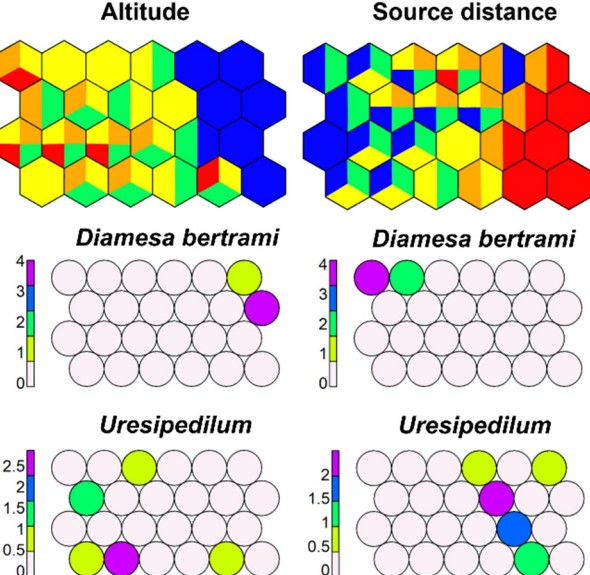

**Figure 20.** Supervised map of sites and two morphotaxa clustered, respectively, using altitude (in m a.s.l.) and source distance (in km) as external variable. Altitudinal range: Blue, >1000 m; green, 500–1000 m; yellow, 100–500 m; orange, 50–100 m; red, <50 m. Source distance range: Blue, <1 km; green, 1–10 km; yellow, 10–50 km; orange, 50–100 km; red, >100 km.

Results from altitude and source distance are compatible with a response of morphotaxa to water temperature, inversely correlated with altitude and directly with source distance.

Results of SSOM carried out with $O_2$ concentrations and $O_2$ % saturation emphasized that response of morphotaxa to oxygen depletion is wide. Few morphotaxa (e.g., *Pagastiella*) are sensitive, while some are tolerant (*C. bicinctus*, *C. plumosus* gr., *Cryptochironomus*) (Figure 21).

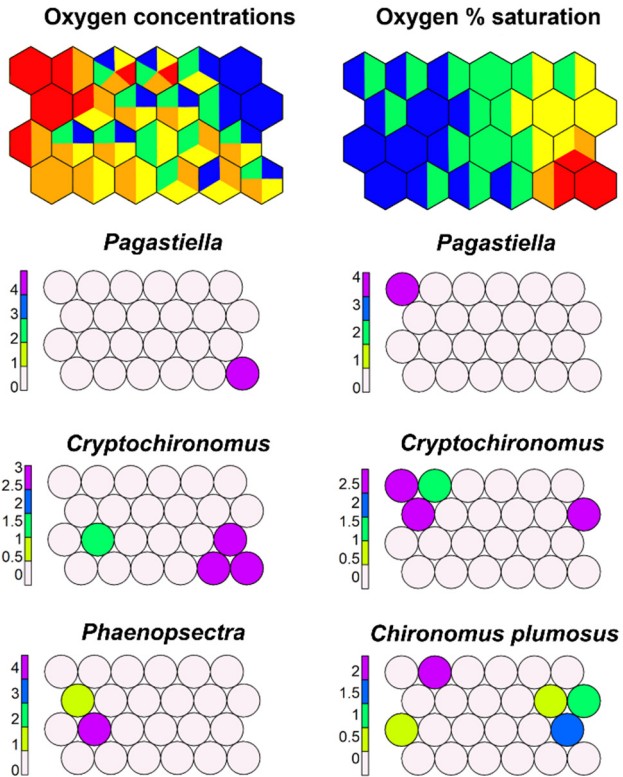

**Figure 21.** Supervised map of sites and three morphotaxa clustered, respectively, using dissolved oxygen (in $mg^{-1}$) and oxygen % saturation as external variable. Oxygen range: Blue, >11 $mg^{-1}$; green, 10–11 $mg^{-1}$; yellow, 9–10 $mg^{-1}$; orange, 5–9 $mg^{-1}$; red, <5 $mg^{-1}$. Oxygen % saturation range: Blue, >100%; green, 80–100%; yellow, 60–80%; orange, 40–60%; red, <40%.

A clear response to current velocity was not observable probably because of the lack of data, with only 136 samples available (Table 1). Maps of response of all the other morphotaxa are deposited at the University of Milan and will not be discussed here.

## 4. Discussion

Chironomids are considered good indicators of ecological status, and at present, the literature concerning the indicator value of the family is copious [42]. Unfortunately, species identification is a long-dated problem and places constraints on the development of chironomids as biological indicators. Discrepancies in the classifications of the three metamorphic stages (larvae, pupae, adults) were only partially resolved with the publication of Holarctic keys to genera [43–45], recently updated for Palaearctic genera at all stages [46] and for Holarctic genera at larval stage [27]. At present, species identifications are still reserved to expert taxonomists, and checklists in hydrobiological works often report only genera with many unidentified species. Freshwater macroinvertebrates studies generally plan the sampling of larvae of insects, but larvae are rarely identifiable to species. The collection of pupal exuviae and adults often accompany larval samples, aiding in species identification [47], but the species lists obtained with this method often include taxa not necessarily associable to the larval stages because pupal exuviae float through the action of water-flow or wind and can disperse over long distances. The use of emergence traps may allow

species identification with the identification of emerging adults [48–50], but this technique is very expensive and could be used only when financial resources are adequate. Another approach is to rear in the laboratory the larvae collected in the field [51], identifying the pupal exuviae obtained or emerging adults, but this method is also time consuming and rather laborious.

Despite this, chironomids are frequently used as indicators in inland waters [52], and species lists characterizing different lake typologies [53–55] or different river zones [49] have been compiled. Many opportunistic species were often observed, *r*-strategists, rapidly invading several habitats when conditions are favourable and rapidly disappearing when favourable conditions cease, so the assignment of an indicator value to a species is often problematic. Examples of opportunistic species are *P. skirwithensis* and *P. rufiventris*. These two species characterize cold mountain and lowland waters, respectively, but they colonize several habitats, such as lake littorals, springs (krenal), and rivers (rhithral, potamal). Some species characterize cold waters at high altitudes: *Orthocladius* (*Mesorthocladius) frigidus*, *E. minor*, and *Pseudodiamesa branicki* [30,32]. Other species, such as *M. atrofasciata*, *I. sylvestris*, and *P. choreus,* are widespread and found in almost all habitats.

Few species are cited as restricted to a single habitat, such as *Cricotopus fuscus* [7] in krenal *E. claripennis* and *Rheocricotopus dispar* [49] in rhithral, *Robackia* and *Harnischia* [2] in sandy substrates in potamal, and *Micropsectra contracta* and *Paracladopelma nigritulum* [48] in lake profundals.

In the present analysis, USOM confirmed that many species are opportunistic, being present in different habitats, and few taxa seemed to be restricted to one habitat only, so it is better to speak about preference more than exclusivism. Some species are strictly restricted to-low temperature waters, such as glacial and cold springs, while others are restricted to running waters, with moderate water velocity (rhithral), and few species seem confined to the lower course of streams (potamal). The separation between lotic and lentic species is generally evident, but there are species found both in lentic and lotic waters (Figure 11).

Interactions between variables may differ in different ecosystems, so one questions the ambition to prepare a table giving indicator weights for different variables for each species [56]. As a result, the input data used to perform calculations deeply affected the attempt to give detailed information with optimum values and ranges of tolerance for each species [57]. Using restricted databases, many habitats characterized by different water temperature, conductivity, acidity, water quality, and velocity can separate different species assemblages, but the same taxa can show tolerance to different factors in different ecosystems. It was often observed that tolerant taxa prevail in pools and sensitive taxa in riffles, but in particular situations, sensitive taxa were found in pools and not in riffles [22].

In the introduction, it was pointed out that identification limited to genus is not sufficient to characterize samples, but the identification extended to species is generally impossible, so a solution must be an intermediate level, that is, the identification of groups of species (morphotaxa) within a genus. The species are not separable within the group, but the morphotaxa are well separable from each other. Pooling different species into a single morphotaxon is critical and requires careful investigation. Often, morphological characters useful to separate larval groups exist but require much effort, so a compromise is necessary between the need to have identifiable taxa and the separation of taxonomic units with different ecological needs.

An aid to discovering the ecological niche is the morphological adaptations to environmental conditions often observed in chironomids. Indeed, morphological traits well reflecting the habitat adaptations were found both in larval and pupal stages. This was emphasized since the beginning of chironomid research: some of the most exciting examples are the lengthening of posterior parapods and the reduction of anal setae on procerci in larvae living in fast-running waters (*Diamesa, Eukiefferiella*) [11], the ramification of pupal thoracic horn of pupae living in oxygen-deficient waters [58–60], the development of a fringe of setae on the anal lobe of pupae living in standing waters, and the reduction of anal lobes in pupae living in running waters [11,58].

Biochemical adaptations are also observable, such as the high haemoglobin content in haemolymph and the capability to perform glycolysis in several species [61,62]. The diverse behavioural response, such as different tube building [6], can also account for tolerance of or resistance against different environmental stresses. Many taxa build tubes to avoid predation, with this strategy aiding also in feeding and respiration [63]. Larvae of temperature tolerant species are able to increase undulation frequency with increasing temperature; therefore, the response to water temperature may be mediated by the capacity to undulate within the tube. Unfortunately, biochemical and physiological traits generally are not detectable as well as differences in morphology, but there are some interesting exceptions, such as the red colour of Chironominae and some tribes of Tanypodinae and the shape of the pupal thoracic horn.

A particular condition of water composition in rare cases allowed us to observe differential response of species. For example, a differential response to acidity was observed in Alpine waters with low pH [64]. The selective response of chironomid species to toxic substances was sometimes observed and was a matter of investigation. The community response in ponds subjected to heavy-metals pollution was detected thanks to the appearance of chromosomic aberrations [65] more than in a shift of assemblage composition. Chironomids were reported as avoiding substrates with toxic substance concentration [66] even if food quality of sediments was more important [67]. Tube building has been supposed to be a protection against toxic substances, such as copper sulphate [68]. Species tolerant to low-oxygen concentrations were observed to be also tolerant to toxics [69]. Research on single taxa more than on a study of the whole assemblage was carried out to analyse the response of chironomids to toxic substances using different biomarkers as transcriptional response, e.g., mouthpart deformities [70]. At most, the community response to toxics was studied considering simple microcosms [71].

## 5. Conclusions

Different chironomid species can be pooled into a single morphotaxon and treated as a single functional unit in analysing ecological responses. If different species belonging to the same morphotaxon have different ecological responses, the assumption on which the concept of morphotaxon relies would be invalidated; otherwise, this assumption will allow greater feasibility to protocols assessing biological water quality, avoiding conclusions based on wrong species identifications. The morphotype concept here proposed appears a good compromise between the conflicting needs of taxonomists and ecologists. A too-fine taxonomic resolution is hard to manage because of the difficulties in identifying the larval stage to species level. As a result, wrong species identifications will give misleading results. A too-low resolution limited to genus or, even worse, to tribe or family level will reduce the indicator capacity.

A morphotaxon often coincides with a genus or a subgenus, but in many cases, a finer division is possible as within the large genera *Eukiefferiella*, *Orthocladius*, *Cricotopus*, and *Chironomus*. Different groups within the genus with different ecological needs will allow a better biomonitoring.

Chironomid species show different sensitivity to several environmental factors with complex interactions; the resulting ecological preferences can then be deeply influenced by the different conditions observed in the different systems. Some environmental variables, such as water temperature and conductivity, are confirmed to be important in determining species selection, and it is surely possible to separate cold-stenothermal species from eurythermal. The same is true for conductivity, acidity [72], and salinity, allowing the separation of steno- from euryhaline species [2,3,11,18,56].

Working with different databases, several factors appear to be dominant in separating species, and this can mislead the interpretation of the species preference. The reason is that a species responds jointly to oxygen, temperature, phosphorous, water velocity, and substratum, that is, a too-great number of factors, and the emphasis on a particular factor may be more bound to the factors included in the model tested than to a true species

preference for a factor. This is critical in testing the validity of biotic indexes in assessing the ecological status of water bodies because the interactions between different environmental factors of natural and anthropogenic origin may give contrasting results [73].

In conclusion, we propose to generically separate tolerant from sensitive taxa and to select sensitive morphotaxa as indicators of water quality. In this way, the present work destroys a great deal and builds little, but the little it builds is very solid and constitutes a good starting point for more in-depth research, which in any case, requires the collection of new data with rigorous experimental protocols.

Future needs are the use of metabarcoding to identify larvae, coupling it with traditional morphological analysis and with the accurate characterization of the environmental variables of the sampled sites [74,75], so the different morphotaxa can be better defined and better related to ecological status.

**Supplementary Materials:** The following supporting information can be downloaded at: https://www.mdpi.com/article/10.3390/w14071014/s1. Supplementary Materials S1: Association of morphotypes with species; Supplementary Materials S2: Input data matrix both in .xlsx format.

**Author Contributions:** Conceptualization, B.R., M.M. and A.B.; methodology, A.B.; software, G.M.; resources, L.M.; visualization, S.Z.; writing—review and editing, B.R., A.B., L.M. and S.Z.; supervision, B.R. All authors have read and agreed to the published version of the manuscript.

**Funding:** This research received no external funding.

**Institutional Review Board Statement:** Not applicable.

**Informed Consent Statement:** Not applicable.

**Data Availability Statement:** All specimens analysed and the related data and all the figures produced are deposited at the University of Milan and can be requested to the first author.

**Conflicts of Interest:** The authors declare no conflict of interest.

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
