# Peer review of "Factors Controlling Morphotaxa Distributions of Diptera Chironomidae in Freshwaters"

_water, doi:10.3390/w14071014_

Round 1

Reviewer 1 Report

Dear authors,

this is a very well written article and well-performed study, both in terms of taxonimic activities and statistical approaches to decipher the data.

The aim of the article is to define the most relevant factors driving Chironomid assemblage composition, considering the assemblage of larvae composed by “morphotaxa” instead of species.

General concept comments

The manuscript is clear, relevant for the field and the methodology used is up to date.

The literature review show the state of art (knowledge)in the field of interest and is well grouped in the introductions. A critical analysis of the working concepts and a presentation of the studies in the research field related to the approached subject is well made.

The introduction of the term “morphotaxon” is a good idea, especially now when the number of researchers (taxonomists) that study classical taxonomy is drastically decreasing.

The experimental design is appropriate to test the hypothesis and the structure of the paper is appropriated to a research article.

We appreciate the large database use for statistical analysis and the type of analysis used.

The results are clear, the figures are very explicit. Conclusions emphasize results obtained by the author.

The author’s contribution is well highlighted and the references are mostly within the last 15 years and does not include an abnormal number of self-citations.

Other observations:

- line 35 - the number of citation in text is [0] or [1]?

- line 50 - for this idea: species known only from fossils - some literature might by introduced (ex: H. Eggermont, D. Verschuren, 2013, CHIRONOMID RECORDS | Africa, Editor(s): Scott A. Elias, Cary J. Mock, Encyclopedia of Quaternary Science (Second Edition), Elsevier, 361-372, ISBN 9780444536426, https://doi.org/10.1016/B978-0-444-53643-3.00272-7, or others).

- line 148-149: "1- [27-29] for all subfamilies, 2- [30, 31] for Diamesinae and

Prodiamesinae, 3- [31, 32] for Orthocladiinae, 7- [33] for Chironominae."- why 7 and not 4?

- line 225-228 - in the figure 1, cell 18 is used MED instand of ME?

- line 232 - "D. latitarsis and D. zernyi in cells 13 and 19 (Figure 2) were clustered in cells 13, 19 and 20" - is necessarily to cut "in cells 13 and 19 (Figure 2)"

- line 334-339 - "brackish waters (B, pink)" is orange in the figure 12, not pink

- line 368-370 "others such as Mesorthocladius, D. latitarsis, D. bertrami preferred low conductivity waters (Figure 16)." - D. latitarsis, D. bertrami are not present in the figure 16, it would be advisable to cite the additional material - supplementary materials -from University of Milano

- line 379-380 "but C. thummi type was the group most tolerant to high TP, Heterotrissocladius, Pagastiella, Corynoneura and Psectrocladius preferred sites with low TP (Figure 17)." - C. thummi, Corynoneura and Psectrocladius are not present in the figure 16, it would be advisable to cite the additional material - supplementary materials - from University of Milano

- line 428 - "Figure 21. supervised map of sites and 3 morphotaxa clustered" are 4 morphotaxa in figure 21

- The titles of the figures must be written in capital letters

Reviewer 2 Report

An admirable paper that needs some English language adjustment as given in the edited e mail sent to the editor.
